# Assessing the Relationship of Different Levels of Pain to the Health Status of Long-Term Breast Cancer Survivors: A Cross-Sectional Study

**DOI:** 10.3390/life15020177

**Published:** 2025-01-25

**Authors:** Francisco Álvarez-Salvago, Maria Figueroa-Mayordomo, Cristina Molina-García, Clara Pujol-Fuentes, Sandra Atienzar-Aroca, Manuel de Diego-Moreno, Jose Medina-Luque

**Affiliations:** 1Department of Physiotherapy, Faculty of Health Sciences, European University of Valencia, 46010 Valencia, Spain; salvagofran@gmail.com (F.Á.-S.); clara.pujol@universidadeuropea.es (C.P.-F.); 2Department of Health Sciences, Faculty of Health Sciences, University of Jaén, 23071 Jaén, Spain; manueldediego7@gmail.com; 3Faculty of Physiotherapy, Podiatry and Occupational Therapy, Catholic University San Antonio-UCAM, 30107 Murcia, Spain; cmolina799@ucam.edu; 4Department of Dentistry, Faculty of Health Sciences, European University of Valencia, 46010 Valencia, Spain; sandra.atienza@universidadeuropea.es; 5Translational Brain Research, German Center for Neurodegenerative Diseases DZNE, 37075 Munich, Germany; jose.med.luque@gmail.com

**Keywords:** long-term survivorship, pain, breast cancer, quality of life, rehabilitation

## Abstract

**Purpose:** This study investigated the relationship between different pain levels in the affected arm and health status in long-term breast cancer survivors (LTBCSs) and identified predictors of pain at this stage of long-term survivorship. **Methods:** A cross-sectional study of 80 participants categorized LTBCSs by pain levels in the affected arm into three groups: no pain (0–0.99), mild pain (1–3.99), and moderate to severe pain (4–10). Variables assessed at least 5 years since diagnosis include pain in the non-affected arm, pain interference, cancer-related fatigue (CRF), physical activity (PA) level, fitness condition, mood state, and health-related quality of life (HRQoL). **Results:** A total of 36.25% of LTBCSs have no pain, 30% have mild pain, and 33.75% have moderate to severe pain. Furthermore, pain presence was associated with increased pain in the non-affected arm, pain interference, CRF, mood disturbances, and physical inactivity, as well as a decreased HRQoL (all *p* < 0.05). Regression analysis found “upset by hair loss”, CRF “affective domain”, “dyspnea”, and “alcohol consumption” as significant predictors of higher levels of pain in the affected arm (r^2^ adjusted = 0.646). **Conclusions:** A total of 63.75% of LTBCSs continue to experience mild to moderate to severe pain in the affected arm, negatively impacting their physical, mental, and emotional health status, with increased pain severity ≥5 years beyond cancer diagnosis. “Upset by hair loss”, CRF “affective domain”, “dyspnea”, and alcohol consumption collectively explain 64.6% of the affected-arm pain level in LTBCSs.

## 1. Introduction

Breast cancer (BC) is one of the most commonly diagnosed cancers within young and adult women, with more than 2 million new cases each year worldwide [1,2]. Despite its striking impact on the population, a large amount of the literature has highlighted that a rising number of women diagnosed with BC are expected to survive up to 5 years after cancer diagnosis, becoming long-term breast cancer survivors (LTBCSs) [3]. The implementation of early screening, detection, and advancements in BC treatments have successfully increased patients’ life expectancy, leading to a decrease in mortality rates over the past few decades [4]. Thus, with improved long-term diagnostics, health-related quality of life (HRQoL) has become a major concern in LTBCS patients, as the majority of them experience physical, mental, and emotional side effects. These include pain, cancer-related fatigue (CRF), and anxiety, which often persist in BC patients even many years beyond cancer diagnosis [5,6,7,8].

Pain has been identified as one of the main consequences affecting HRQoL in BC survivors, and it is associated with CRF, lower fitness condition and physical activity (PA) levels, and depression and anxiety in LTBCSs [9,10,11]. The prevalence among patients is significant, as chronic and persistent pain have unfortunately been reported in approximately 30–60% of LTBCs patients [12,13]. Although previous studies on BC survivors have determined certain risk factors such as age, surgery, or genetics for the persistence of pain [14], it remains unclear how those factors may contribute to or be correlated with the development of pain in LTBCSs. While several analyses have studied the impact that several factors may have on those patients, there is no consensus on which may be the cause of the persistence of pain [15,16]. In addition, most 5-year post-diagnosis follow-up studies have evaluated the factors influencing persistent pain without considering the physical or psychological characteristics associated with it in these patients.

Given the prevalence of pain among LTBCSs and its potential impact on their lifespan, it is crucial to keep on elucidating which factors may be contributing to the persistence of pain in order to enhance prevention strategies and rehabilitation programs. Therefore, the aim of this study was to analyze the possible relationship between different pain levels in the affected arm and health status in LTBCSs. The second aim was to identify predictors of pain at this stage of long-term survivorship.

## 2. Materials and Methods

### 2.1. Design and Participants

A cross-sectional study was conducted in 2022 at the Sport and Health Joint University Institute (iMUDS), comprising 80 LTBCSs identified from the oncology service at the University Hospital Complex of Granada.

The required sample size was calculated using G*Power (Version 3.1.9.7) for a comparison of three independent groups. Assuming a medium effect size (f = 0.25), an alpha level of 0.05, and a power of 0.80, the required sample size per group is 24. Thus, the total sample size required is 72 participants. However, to account for potential imbalances and to ensure sufficient power, we recruited a total of 80 participants, which provides a sufficient sample size for our analyses. The flow diagram detailing the study participants can be found in Appendix A.

Interested participants were initially contacted by phone, provided with detailed study information, and invited to address any questions. Inclusion criteria for LTBCSs included the following: (1) age ≥ 18 years and (2) a minimum of 5 years since diagnosis of stage I–IIIa BC. Exclusion from the study encompassed individuals who were unable to read or understand the assessments. Subsequently, a research team physiotherapist obtained written informed consent in person. The assessment session, lasting approximately 45 min, followed.

Based on cut-off points from previous studies, which categorized pain as mild (0–3), moderate (4–6), and severe (≥7) [17], we refined the classification to create three homogeneous groups. Specifically, the original mild pain category (0–3) was subdivided into two groups: no pain (0–0.99) and mild pain (1–3.99). Additionally, the moderate and severe pain categories were combined into a single group, labeled “moderate/severe pain”, as seen in previous studies [18]. Consequently, participants were categorized into three groups according to their VAS scores: no pain (0–0.99), mild pain (1–3.99), and moderate/severe pain (4–10).

The Biomedical Research Ethics Committee of Granada (CEIm) (1038-N-16 I.P/07/26/2018) granted ethical approval for this study, adhering to the principles of the Declaration of Helsinki (14/2017) [19].

### 2.2. Variables

#### 2.2.1. Demographic, Clinical, and Medical Data Collection

Data collection was performed through structured personal interviews utilizing a tailored questionnaire. This instrument gathered comprehensive information on several variables, including clinical characteristics related to the illness and sociodemographic factors such as age, marital status, educational level, and employment status. Specific data points of clinical characteristics included time since diagnosis, tumor stage, history of tobacco and alcohol use, family history of BC, menopausal status if applicable, treatment type, surgical interventions, current medication use, presence of metastasis or recurrence, and participation in psychological or physiotherapy services.

#### 2.2.2. Pain Measures

The VAS, with an intra-class correlation coefficient (ICC) of 0.97 [20], is a 10 cm linear scale used to measure subjective pain intensity, where 0 represents “no pain” and 10 indicates “the worst imaginable pain”. Participants were asked to rate their current pain levels in both the affected and non-affected arms. For cases of bilateral BC, where both arms are affected, the “affected arm” was defined as the one more impacted by the disease or its treatment. This designation was based on the arm that underwent more severe interventions, such as intensive surgery or radiotherapy, as these treatments may result in greater pain, lymphedema, or other conditions with asymmetrical manifestations. The determination of the affected and non-affected arms considered the following criteria: (1) the extent of surgical invasiveness, (2) the presence of lymphedema or other post-surgical complications, and (3) the patient’s self-perception of pain comparing both arms.

The Brief Pain Inventory (BPI) short form, with a Cronbach’s alpha ranging from 0.87 to 0.89 [21], was utilized to evaluate pain severity (intensity) using four items and its impact on daily functioning (interference) using seven items. For statistical analysis, only the pain interference values were considered.

#### 2.2.3. Cancer-Related Fatigue

The Piper Fatigue Scale (PFS), with a Cronbach’s alpha of 0.86 [22], assesses CRF across four domains: behavioral/severity, affective, sensory, and cognitive/mood. This 22-item scale provides a comprehensive total score, with higher values reflecting a greater CRF [23,24].

After consideration of results published in previous publications, two cut-off score models (A and B) were deemed effective in distinguishing fatigue categories. These models were Model A (0 = none, 1–3 = mild, 4–6 = moderate, 7–10 = severe) and Model B (0 = none, 1–2 = mild, 3–5 = moderate, 6–10 = severe) [23,24]. Furthermore, moderate fatigue, regardless of the model applied, has previously been considered as clinically significant and therefore warranting further attention [25]. Consequently, participants’ CRF levels were classified using both models to explore potential differences in the results based on these established categorizations.

#### 2.2.4. Physical Activity Level

The Minnesota Leisure Time Physical Activity (MLTPA) questionnaire, with an ICC of 0.95 [26], was employed to assess the average hours and frequency dedicated to PA over the previous week with a list of clearly defined physical activities. Energy expenditure was estimated by multiplying the reported time (in hours per week) spent on each activity mentioned in the questionnaire by its corresponding Metabolic Equivalent of Task (MET) value [27], reflecting the activity’s energy cost.

#### 2.2.5. Fitness Condition

The International Fitness Scale (IFIS), with a Cronbach’s alpha of 0.80 [28], assesses self-perceived fitness through five key questions that evaluate overall physical fitness as well as specific components, including cardiorespiratory fitness, muscular strength, speed/agility, and flexibility, in comparison to peers. Outcomes are rated on a 5-point Likert scale, ranging from 1 (very poor) to 5 (very good).

#### 2.2.6. Mood State

The Scale for Mood Assessment (EVEA), with a Cronbach’s alpha ranging from 0.88 to 0.93, demonstrates good reliability in assessing four mood dimensions [29]. The scale consists of 16 items, each rated on a Likert scale from 0 to 10. The scores for each mood category, namely sadness–depression, anxiety, anger–hostility, and happiness, are calculated as the mean of the corresponding items.

#### 2.2.7. Health-Related Quality of Life

Two questionnaires were used to assess HRQoL, specifically the EORTC QLQ-C30 version 3.0 and its BC module QLQ-BR23, with a Cronbach’s alpha ranging from 0.46 to 0.94 [30,31]. Patient outcomes for each question are rated on a 4-point scale (1 = not at all, 4 = very much) and, subsequently, linearly transformed to range from 0 to 100. As for the interpretation of results, higher scores for the functional and global HRQoL scales indicate better health, while higher scores for the symptom scales indicate more symptom burden. Additionally, a summary score for the QLQ-C30 was calculated, combining 13 QLQ-C30 scale and item scores, excluding global QoL and financial impact. For this summary score, higher scores indicated a better HRQoL [32].

#### 2.2.8. Statistical Analysis

The Statistical Package for the Social Sciences (IBM SPSS Statistic for Windows, Armonk, NY, USA, version 27.0) was used for data analysis, and the significance level was set at (*p* < 0.05) a 95% confidence interval (CI).

Normality was tested for all variables using the Kolmogorov–Smirnov test (*p* > 0.05). As for continuous variables, for normally distributed variables, ANOVA was used to compare the three groups: no pain (0–0.99), mild pain (1–3.99), and moderate/severe pain (4–10). For non-normally distributed variables, the Kruskal–Wallis test was used, followed by pairwise comparisons using Mann–Whitney U tests. With regards to both categorical and ordinal variables, Chi-square tests were used to compare categorical variables. Furthermore, between-group effect sizes for continuous variables were calculated using Cohen’s *d*: negligible (*d* = 0–0.19), small (*d* = 0.2–0.49), moderate (*d* = 0.5–0.79), large (*d* = 0.8–1.19), and very large (*d* = ≥1.20) [33].

A Spearman correlation analysis was performed to explore the association between pain in the affected arm, assessed using the VAS, and other variables included in this study. Furthermore, a stepwise multiple regression analysis was applied to determine the factors contributing to the variability in pain in the affected arm. Inclusion of variables in the multiple regression model required two conditions to be met: (1) a statistically significant correlation with the dependent variable, and (2) an inter-variable correlation coefficient below 0.70 among independent variables to avoid collinearity issues [34,35,36]. A forward selection approach was utilized to introduce significant predictors into the regression model one at a time, based on their degree of association with the dependent variable. At each step, the statistical significance of the regression outcomes was evaluated, and the standardized β coefficients for each variable in the final model were computed. Logarithmic and square root transformations were applied to non-normally distributed variables, as assessed by the Kolmogorov–Smirnov test, to meet linear regression assumptions.

## 3. Results

### 3.1. Demographic and Clinical Characteristics

No significant differences were observed between groups regarding the demographic and clinical characteristics of the 80 participants based on the level of pain in the affected arm among LTBCSs. Using adapted established criteria for pain levels [17,18], participants were categorized as experiencing no pain (36.25%), mild pain (30%), or moderate to severe pain (33.75%) in the affected arm.

The mean age for participants with no pain was 51.12 ± 8.74 years, for those with mild pain was 45.00 ± 6.82 years, and for those with moderate to severe pain was 50.51 ± 7.00 years. Among participants with no pain, 31% were on sick leave, 93.1% had undergone both radiotherapy and chemotherapy, and 3.4% had undergone a unilateral mastectomy. Among participants in the mild pain group, 37.5% were on sick leave, 75% had received both treatments, and 41.6% had a unilateral mastectomy. Finally, in the moderate to severe pain group, 44.4% were on sick leave, 81.5% had received radiotherapy and chemotherapy, and 40.7% had undergone a unilateral mastectomy. Further details of the demographic and clinical characteristics are presented in Table 1.

### 3.2. Pain

Analysis of VAS scores for the non-affected arm revealed significant differences between groups. Comparisons between the no pain and moderate to severe pain groups, as well as the mild pain and moderate to severe pain groups, showed significantly higher scores in the moderate to severe pain group, with very large effect sizes (U = 157.50 and 107.00; *p* = 0.01, *d* = 1.34 and 1.39, respectively). No significant differences were observed when comparing the no pain to the mild pain group.

Additionally, analysis of BPI interference scores indicated significantly higher values in the moderate to severe pain group compared to both the mild pain and no pain groups (U = 191.00 and 185.00; *p* < 0.01 and 0.01, respectively). Effect sizes ranged from medium to large, respectively (*d* = 0.59 and 1.17), as shown in Table 2.

### 3.3. Cancer-Related Fatigue

When analyzing the PFS domains, comparisons between the no pain and mild pain groups revealed significantly higher values across all domains in the mild pain group—“behavioral/severity” (U = 229.00; *p* = 0.05), “affective” (U = 162.00; *p* < 0.01), “sensory” (U = 157.50; *p* < 0.01), “cognitive/mood” (U = 204.50; *p* = 0.01), and “total fatigue score” (U = 179.00; *p* < 0.01)—with effect sizes ranging from medium to large (*d* ranging from 0.56 to 1.00). Similarly, comparisons between the no pain and moderate to severe pain groups showed significant increases in all domains for the moderate to severe pain group—“behavioral/severity” (U = 236.00; *p <* 0.01), “affective” (U = 181.50; *p* < 0.01), “sensory” (U = 176.50; *p* < 0.01), “cognitive/mood” (U = 190.00; *p* < 0.01), and “total fatigue score” (U = 164.00; *p* < 0.01)—with effect sizes ranging from large to very large (*d* = 0.84 to 1.23). In contrast, comparisons between the mild pain and moderate to severe pain groups demonstrated that only the “cognitive/mood” domain was significantly higher in the moderate to severe pain group, with a medium effect size (U = 177.00; *p* = 0.04; *d* = 0.63) (Table 2).

For cut-score type A, significant differences were observed when comparing the no pain group to both the mild pain and moderate to severe pain groups (*p* < 0.01). In this regard, while only 17.1% of LTBCSs reported “moderate” to “severe” CRF levels in the no pain group, the mild pain and moderate to severe pain groups showed “moderate” to “severe” CRF levels, reaching 54.1% and 62.9%, respectively (Table 2). Statistical analysis of cut-score type B revealed only significant differences between the no pain and moderate to severe pain groups (*p* < 0.01). The no pain group had 24% of participants reporting “moderate” to “severe” CRF, compared to 70.3% in the moderate to severe pain group. No significant differences in CRF scores were observed when comparing the no pain group to the mild pain group or the mild pain group to the moderate to severe pain group (Table 2).

### 3.4. Physical Activity Level

Analysis of between-group differences in MLTPA scores revealed a significant difference between the no pain and moderate to severe pain groups. The moderate to severe pain group exhibited a higher proportion of “inactive” participants (40.7%), whereas the no pain group showed a lower proportion of “inactive” participants (13.7%) (*p* = 0.05) (Table 2).

### 3.5. Fitness Condition

The analysis of fitness condition, as measured by the IFIS, showed significant differences across groups. Comparisons between the no pain and mild pain groups (U = 147.00 to 214.50; *p* < 0.01 to 0.02), as well as the no pain and moderate to severe pain groups (U = 167.50 to 343.50; *p* < 0.01 to 0.02), revealed significantly higher scores for the no pain group across all domains in both comparisons, with effect sizes ranging from medium to very large (*d* = 0.64 to 1.35). For a visual representation of these differences, refer to Figure 1 and Appendix A.

### 3.6. Mood State

Comparisons of mood states between the no pain and mild pain groups, as assessed by the EVEA, revealed significantly higher values in the mild pain group only for the domain “sadness–depression” (U = 223.00; *p* = 0.04, *d* = 0.36). In contrast, all mood state domains, except for “happiness”, were seen as significantly higher for the moderate to severe pain group when comparing it to both the no pain and mild pain groups (U = 222.50 to 248.50; *p* < 0.01, *d* = 0.82 to 0.93 and U = 159.00 to 177.50; *p =* 0.01 to 0.04, *d* = 0.59 to 0.73, respectively). Additional details are provided in Table 2.

### 3.7. Health-Related Quality of Life

In relation to the QLQ-C30 and BR23, when comparing no pain to mild pain, higher significant differences were observed only in “role” and “sexual functioning”, and “body image”, as well as lower levels of “nausea and vomiting”, “dyspnea”, “arm symptoms”, and “upset by hair loss” (U = 169.50 to 226.00; *p* = 0.01 to 0.04, *d* = 0.56 to 0.62). Interestingly, the mild pain group exhibited lower “systemic therapy side effect” values with respect to the no pain group (U = 228.00; *p* = 0.01; *d* = 0.78) (Table 3).

With reference to no pain compared to moderate to severe pain, statistical analysis showed significantly higher functioning values in both questionnaires for the no pain group except for “physical functioning”, “sexual enjoyment”, and “future perspectives”, as well as higher values in the summary score (U = 143.50 to 300.00; *p* < 0.01 to 0.03; *d* = 1.05 to 2.85). However, continuing with this comparison, the moderate to severe pain group exhibited lower “systemic therapy side effect” values with respect to the no pain group (U = 296.50; *p* = 0.03; *d* = 0.61). As for symptom scales, it could also be observed in both questionnaires how, in this group comparison, the moderate to severe pain group exhibited higher “fatigue”, “nausea and vomiting”, “dyspnea”, “insomnia”, “diarrhea”, “financial difficulties”, “breast symptoms”, “arm symptoms”, and “upset by hair loss” with respect to the no pain group (U = 68.00 to 226.00; *p* < 0.01 to 0.02; *d* = 0.61 to 1.42) (Table 3).

When comparing the mild pain and moderate to severe pain groups, significantly higher values were only observed in a few functioning scales, namely “role” and “cognitive” and “social functioning”, as well as the summary score (U = 105.50 to 158.00; *p* < 0.01 to 0.01; *d* = 0.53 to 1.04). On the contrary, the moderate to severe group showed higher levels of symptoms for “fatigue”, “dyspnea”, “insomnia”, “diarrhea”, “breast symptoms”, “arm symptoms”, and “upset by hair loss” compared to the mild pain group (U = 87.00 to 180.50; *p* < 0.01 to 0.04; *d* = 0.27 to 1.46). Additional details are provided in Table 3.

### 3.8. Correlation Analysis

The Spearman’s correlation analysis showed significant positive correlations between pain in the affected arm and the following variables: “type of surgery”; PFS: “behavioral/severity”, “affective”, “sensory”, “cognitive/mood”, and “total fatigue”; QLQ-C30: “nausea and vomiting”, “pain”, “dyspnea”, “insomnia”, “appetite loss”, “diarrhea”, “financial difficulties”, and “global health”; QLQ-BR23: “breast symptoms”, “arm symptoms”, and “upset by hair loss”; VAS: “pain in the non-affected arm”; BPI: “pain interference”; EVEA: “sadness–depression”, “anxiety”, and “anger–hostility” (*r* = 0.222 to 0.689; *p* < 0.01 to 0.05). Significant negative correlations were observed between pain in the affected arm and the following variables: “alcohol consumption” and “type of treatment”; QLQ-C30: “role functioning”, “emotional functioning”, “cognitive functioning”, and “social functioning”; QLQ-BR23: “body image”, “sexual functioning”, and “systemic therapy side effects”; IFIS: “general physical fitness”, “cardiorespiratory fitness”, “muscular strength”, “speed/agility”, and “flexibility” (*r* = −0.221 to −0.512; *p* < 0.01 to 0.05). The results are represented in Figure 2.

### 3.9. Multiple Regression Analysis

The final regression model identified the following variables as significant predictors of pain in the affected arm: “upset by hair loss” from the QLQ-C30, CRF “affective domain” from the PFS, “dyspnea” from the QLQ-C30, and “alcohol consumption”. Together, these factors explained 64.6% of the variance in pain levels (r^2^ adjusted = 0.646; *p* =< 0.01 to 0.05) in individuals who were ≥5 years beyond cancer diagnosis. Detailed results of the multiple regression analysis are provided in Table 4.

## 4. Discussion

The purpose of this study was to investigate the relationship between different pain levels in the affected arm and the health status of LTBCSs and to identify predictors of pain at this stage of long-term survivorship. The main findings of this study indicated that ≥5 years beyond cancer diagnosis, 36.25% of LTBCSs have no pain, 30% have mild pain, and 33.75% have moderate to severe pain. Furthermore, higher pain severity was associated with increased pain in the non-affected arm, pain interference, CRF, mood disturbances, and physical inactivity, as well as a decreased HRQoL. Notably, 64.6% of affected-arm pain level variance was explained by “upset by hair loss”, CRF “affective domain”, “dyspnea”, and alcohol consumption.

Regarding the prevalence of pain, our results show that pain persists in 63.75% of our LTBCSs, with 33.75% experiencing moderate to severe pain ≥5 years after BC diagnosis. These findings are in line with, and even exceed, the percentages reported by other studies on the prevalence of moderate to severe pain in women in the long-term survival phase [5,37,38,39]. Similarly, it was observed that in our LTBCSs, higher pain in the affected arm was not only associated with the onset of pain in the non-affected arm, but also with greater interference in performing daily activities. This is consistent with previous research highlighting how chronic pain in the affected arm can lead to functional compensation, overuse of the contralateral arm, and neurological sensitization, which increase bilateral pain [40,41]. However, the fact that these studies did not specifically focus on survivors ≥5 years, and that our findings suggest these dynamics may persist long-term, underscores the need for further research on these aspects during the long-term survival phase.

With respect to CRF, our analysis of CRF domains in LTBCSs revealed significant differences between groups classified by pain intensity. LTBCSs in the mild pain group reported significantly higher CRF scores across all domains compared to those with no pain. Similarly, those in the moderate to severe pain group demonstrated even greater levels of CRF across all domains compared to the no pain group. These findings align with previous studies showing that higher pain levels in LTBCSs are related to greater CRF and functional impairment [42,43]. Interestingly, the only domain that differed significantly between the mild and moderate to severe pain groups was CRF “cognitive/mood”, with the moderate to severe pain group reporting higher scores, suggesting that cognitive impacts may be more pronounced in this subgroup.

On the other hand, our results also revealed significant differences in the moderate to severe pain group, which presented higher percentages of “moderate” to “severe” CRF for cut-score A (62.9%) compared to the no pain group (17.1%) and the mild pain group (54.1%). This finding was also significant between the moderate to severe pain group (70.3%) and the no pain group (24%) for cut-score B. Given that moderate CRF is linked with the need for clinical intervention [25], these findings underscore the importance of early identification and management of CRF in LTBCSs, especially in those experiencing moderate to severe pain.

As for PA and fitness condition, pain has been consistently identified as a limiting factor for PA in LTBCSs, with evidence indicating that pain, particularly in the affected arm, may lead to reduced activity levels through mechanisms such as discomfort, fear of exacerbating symptoms, or functional limitations [44]. Studies focusing on BC survivors beyond the acute treatment phase have reported similar associations between pain and physical inactivity [40,45,46], although evidence beyond 5 years remains sparse [44]. Our results further demonstrate this relationship, as LTBCSs experiencing moderate to severe pain exhibited significantly higher proportions of inactivity compared to those with no pain (40.7% vs. 15.2%). In terms of perceived physical fitness, comparisons revealed that participants in the mild pain and moderate to severe pain groups consistently reported poorer scores across all domains compared to the no pain group, suggesting a potential link between higher pain levels and poorer self-perceptions of fitness. This observation is consistent with previous studies indicating that pain adversely affects not only physical performance but also subjective evaluations of physical capabilities. However, most of these findings are limited to earlier stages of survivorship [16,44,45]. Given the implications of both inactivity and diminished perceived fitness for long-term health outcomes [47], these findings underscore the critical need for targeted interventions addressing pain management and physical rehabilitation in LTBCSs, particularly in those with moderate to severe pain.

In the case of mood state, our results further reveal that LTBCSs experiencing moderate to severe pain reported significantly poorer mood states, including higher levels of sadness–depression, anxiety, and anger–hostility compared to both the no pain and mild pain groups. This pattern aligns with previous studies that suggest chronic pain, especially in the context of cancer survivorship, can exacerbate negative emotional states [16,48]. The relationship between pain and mood disturbance can be attributed to various mechanisms, such as the increased activation of inflammatory pathways that affect brain regions responsible for mood regulation. These neurobiological changes, together with physical limitations caused by pain, contribute to a cycle of emotional distress and decreased HRQoL in LTBCSs [13,37,49]. Given the emotional burden associated with persistent pain, addressing both the physical and psychological aspects of pain in LTBCSs is crucial for improving their overall well-being.

Considering HRQoL, our results reflect a trend in which LTBCSs with higher pain intensity seem to report worse HRQoL, both in functioning symptom domains, as assessed by the QLQ-C30 and QLQ-BR23. In particular, the moderate to severe pain group showed higher levels of symptoms and both lower levels of functioning and lower summary score compared to the no pain and mild pain groups. These results align with previous studies, except for the summary score, which was not examined in those investigations, highlighting that persistent pain can amplify symptom burden and restrict physical, social, and emotional functioning in LTBCSs [16,37,50]. Interestingly, the moderate to severe pain group also reported significantly lower values for “systemic therapy side effects” compared to both the no pain and mild pain groups. These “systemic therapy side effects”, as assessed by the QLQ-BR23, include symptoms such as dry mouth, altered taste, eye pain, hair loss, feeling sick, hot flashes, and headaches. We believe that one possible explanation for this observation could be that individuals experiencing more pain may have developed a greater tolerance to or adaptation to systemic side effects over time, particularly if the pain has become more chronic. Another hypothesis is that the presence of more severe pain might lead to a greater focus on pain management, which could overshadow or reduce the perception of other symptoms. Thus, our finding warrants further investigation, as understanding the interplay between chronic pain and the perception of systemic side effects could have implications for treatment and symptom management in LTBCSs.

In discussing the results of our correlation and regression analysis, our findings align with the existing literature regarding LTBCSs. Specifically, we observed significant positive correlations between pain in the affected arm and a variety of health and psychological variables, such as the aggressiveness of surgery, CRF, dyspnea, insomnia, and emotional distress (e.g., sadness–depression, anxiety, anger–hostility, and upset by hair loss). This mirrors previous research that has demonstrated that chronic pain in LTBCSs is commonly linked to increased symptom burden and emotional distress, which can severely impact overall health and well-being [5,16,37,38,51,52]. Similarly, the significant negative correlations we observed between pain and physical, role, and emotional functioning, as well as fitness condition and alcohol consumption, are consistent with other studies that have highlighted the detrimental effects of persistent pain on daily functioning and HRQoL [5,16,37,38,51,52].

In terms of the regression analysis, and when compared to other studies, the percentage of variance explained by our model (64.6%) exceeds the ranges proposed by the few studies that have explored possible predictors of pain in LTBCSs, ranging from 30% to 60% [16,37,38,51,52]. This underscores the variability inherent in research methodologies and emphasizes the need for further investigation. Exploring broader and more standardized predictor sets may enhance the understanding and management of long-term pain in this population.

In this sense, we identified four significant predictors (“upset by hair loss”, CRF “affective domain”, “dyspnea”, and alcohol consumption) explaining 64.6% of the variance in pain levels, which contributes to the growing body of research on long-term pain management in LTBCSs. Previous studies have shown that various factors, including emotional distress, CRF, body image concerns, and physical symptoms like dyspnea, are key predictors of long-term pain in cancer survivors [52]. The role of “upset by hair loss” as a predictor of pain intensity is particularly noteworthy. This finding aligns with research indicating that psychological factors related to body image, such as the distress caused by hair loss, can exacerbate pain perception and emotional distress in LTBCSs [53]. Additionally, another interesting finding in our study was the role of alcohol consumption as another predictor, with higher pain intensity correlating with lower alcohol use. This observation aligns with research suggesting that while alcohol may initially be used as a self-management strategy for pain relief, its chronic use can exacerbate pain perception through mechanisms such as neuroinflammation, leading to reduced consumption among individuals experiencing persistent pain [54,55]. Furthermore, in LTBCSs, reduced alcohol intake may reflect heightened health awareness and lifestyle adjustments following their cancer diagnosis. Survivors often adopt healthier behaviors to mitigate recurrence risks and manage symptoms, which could explain the lower consumption levels in individuals with higher pain intensities [54]. These findings underscore the complex interplay between behavioral coping mechanisms and physiological responses in pain management among long-term cancer survivors.

Finally, this study has various limitations that merit attention. Firstly, the approach to categorizing participants into no pain, mild pain, and moderate to severe pain was adapted from previously established and accepted cut-off points, with modifications to ensure the creation of homogeneous groups and to include a distinct no pain category for participants scoring less than 1 on the VAS scale [17,18], although other cut-off values could have modified our results. However, the inclusion of a distinct no pain category for participants scoring less than 1 on the VAS scale adds value by distinguishing those with no pain (0) from those experiencing mild pain (1–3.99), providing a clearer understanding of pain severity distribution. Secondly, having used more objective variables (although all the tests have been previously validated) could have strengthened our results. Thirdly, the observational design prevents us from drawing causal conclusions. Despite these cited limitations, the key strength of this study lies in its comprehensive approach, exploring the relationship between pain levels and multiple health-related factors in LTBCSs. By identifying significant predictors such as “upset by hair loss”, CRF “affective domain”, “dyspnea”, and alcohol consumption, this study provides valuable insights into the complex nature of pain in LTBCSs. Moreover, the study’s rigorous statistical approach, with a regression model explaining 64.6% of the variance in pain, adds substantial clinical relevance, offering a nuanced understanding of the physical and psychological factors affecting pain in this population. Recognizing the predictors identified in this study can be useful in tailoring treatment plans and developing clinical protocols that address the multifaceted nature of pain management in this population. This could ultimately enhance patient care and improve HRQoL for LTBCSs by considering both physical and emotional health aspects in long-term survivorship.

## 5. Conclusions

In conclusion, even five years post-BC diagnosis, 63.75% of LTBCSs continue to experience mild to moderate to severe pain in the affected arm, which negatively impacts their physical, mental, and emotional health status with increasing pain severity. Moreover, this study provides valuable insights into potential predictors of pain, with “upset by hair loss”, CRF “affective domain”, “dyspnea”, and alcohol consumption collectively explaining 64.6% of its variability in LTBCSs.

## Figures and Tables

**Figure 1 life-15-00177-f001:**
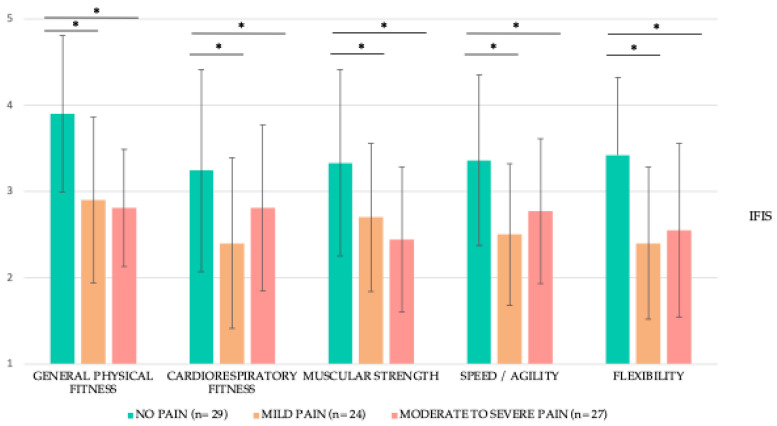
Fitness condition of LTBCSs according to the level of pain in the affected arm expressed as mean ± SD. Abbreviations: IFIS: International Fitness Scale, *n*: sample size, SD: standard deviation. Note: International Fitness Scale values are as follows: (1) very poor, (2) poor, (3) average, (4) good, and (5) very good. Note: The median and inter-quartile range can be found in Appendix A. *p*-values for between-group differences were calculated using the *t* test (Mann–Whitney U test). * *p* ˂ 0.05.

**Figure 2 life-15-00177-f002:**
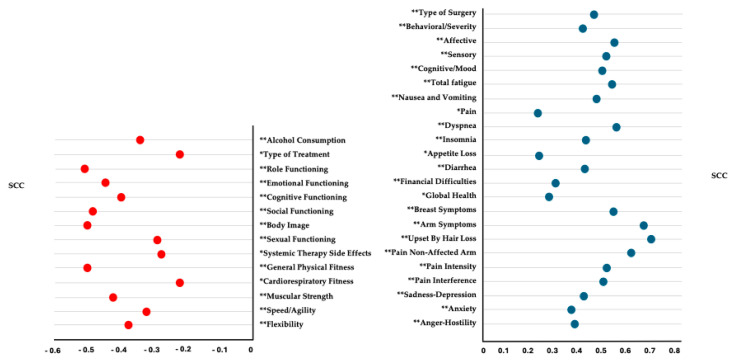
Spearman’s correlation coefficient for pain in the affected arm using the Visual Analogue Scale (VAS). Abbreviations: VAS: Visual Analogue Scale, PFS: Piper Fatigue Scale, IFIS: International Fitness Scale, EVEA: Scale for Mood Assessment, QLQC30: the EORTC Core Quality of Life Questionnaire, QLQBR23: the Breast Cancer-Specific Module, SCC: Spearman’s correlation coefficient. Note: For this analysis, the VAS data as the dependent variable were used in their non-categorized version. * *p* ˂ 0.05/** *p* < 0.01.

**Table 1 life-15-00177-t001:** Demographic, clinical, and medical characteristics of LTBCSs according to the level of pain in the affected arm.

	LTBCSs’ Level of Pain in the Affected Arm			
Characteristics	No Pain (NP)	Mild Pain (MP)	Moderate to Severe Pain (MTSP)	*p*/*x*^2^
	0–0.99 (VAS)	1–3.99 (VAS)	4–10 (VAS)	
	(*n* = 29)	(*n* = 24)	(*n* = 27)	
Mean age ± SD, years	51.12 ± 8.74	45.00 ± 6.82	50.51 ± 7.00	0.16 ^a^
Mean weight ± SD, kg	69.77 ± 9.81	63.85 ± 7.24	67.97 ± 9.58	0.08 ^a^
Mean height ± SD, cm	161.27 ± 6.22	160.51 ± 4.99	159.97 ± 6.51	0.70 ^a^
Mean body mass index ± SD, kg/m^2^	26.84 ± 3.61	24.88 ± 3.34	26.69 ± 4.41	0.16 ^a^
Mean time since diagnosis ± SD, months	87.46 ± 26.30	96.20 ± 31.43	89.88 ± 31.39	0.57 ^a^
Mean time since the first surgery ± SD, months	85.50 ± 26.73	91.70 ± 32.06	87.14 ± 31.99	0.76 ^a^
Marital status, *n* (%)				
Unmarried	5 (17.2)	5 (20.8)	4 (14.8)	0.52 ^b^
Married	20 (68.9)	16 (66.6)	16 (59.3)	
Divorced	2 (6.8)	2 (8.33)	5 (18.5)	
Widowed	2 (6.8)	1 (4.16)	2 (7.4)	
Educational level, *n* (%)				
Primary school	14 (48.2)	7 (29.1)	15 (55.6)	0.18 ^b^
Secondary school	6 (20.6)	5 (20.8)	6 (22.2)	
University	9 (31)	12 (50)	6 (22.2)	
Employment status, *n* (%)				
Housewife	9 (31)	8 (33.3)	8 (29.6)	0.58 ^b^
Currently working	9 (31)	5 (20.8)	3 (11.1)	
Sick leave	9 (31)	9 (37.5)	12 (44.4)	
Retired	2 (6.8)	2 (8.33)	4 (14.8)	
Tumor stage, *n* (%)				
I	9 (31)	7 (29.1)	7 (25.9)	0.86 ^b^
II	15 (51.7)	14 (58.3)	17 (63)	
IIIa	5 (17.2)	3 (12.5)	3 (11.1)	
Dominant side, *n* (%)				
Right-sided	24 (82.7)	20 (83.3)	25 (92.6)	0.62 ^b^
Left-sided	5 (17.2)	4 (16.6)	2 (7.4)	
Tumor location, *n* (%)				
Right side	9 (31)	8 (33.3)	9 (33.3)	0.47 ^b^
Left side	20 (68.9)	13 (54.1)	16 (59.3)	
Bilateral	0 (18.2)	3 (12.5)	2 (7.4)	
Tumor location on dominant side, *n* (%)				
No	13 (44.8)	10 (41.6)	9 (33.3)	0.59 ^b^
Yes	16 (55.1)	14 (58.3)	18 (66.7)	
Tobacco consumption, *n* (%)				
Non-consumption	13 (44.8)	12 (50)	15 (50)	0.76 ^b^
Smoker	6 (20.6)	7 (29.1)	6 (23.8)	
Ex-smoker	10 (34.4)	5 (20.8)	6 (26.3)	
Alcohol consumption, *n* (%)				
Non-consumption	6 (20.6)	7 (29.1)	17 (63)	0.06 ^b^
Monthly	8 (27.5)	6 (25)	5 (18.5)	
Weekly	15 (51.7)	7 (29.1)	4 (14.8)	
Daily	0 (0)	4 (16.6)	1 (3.7)	
Family history of breast cancer, *n* (%)				
No	13 (44.8)	13 (54.1)	13 (48.1)	0.52 ^b^
Yes	16 (55.1)	11 (45.8)	14 (51.9)	
Menopause, *n* (%)				
No	3 (10.3)	6 (25)	4 (14.8)	0.52 ^b^
Yes	26 (89.6)	18 (75)	23 (85.2)	
Type of treatment, *n* (%)				
None	0 (0)	0 (0)	0 (0)	0.17 ^b^
Radiotherapy	0 (0)	0 (0)	3 (11.1)	
Chemotherapy	2 (6.8)	6 (25)	2 (7.4)	
Radiotherapy and chemotherapy	27 (93.1)	18 (75)	22 (81.5)	
Surgery, *n* (%)				
Lumpectomy	10 (34.4)	3 (12.5)	2 (7.4)	0.06 ^b^
Quadrantectomy	17 (58.6)	9 (37.5)	12 (44.4)	
Unilateral mastectomy	1 (3.4)	10 (41.6)	11 (40.7)	
Bilateral mastectomy	1 (3.4)	2 (8.3)	2 (7.4)	
Type of medication, *n* (%)				
None	10 (34.4)	4 (16.6)	4 (14.8)	0.06 ^b^
Tamoxifen	8 (27.5)	14 (58.3)	10 (37)	
Other types	11 (37.9)	6 (25)	13 (48.1)	
Metastasis, *n* (%)				
No	24 (82.7)	18 (75.00)	23 (85.2)	0.59 ^b^
Yes	5 (17.2)	6 (25.00)	4 (14.8)	
Recurrence, *n* (%)				
No	23 (79.3)	16 (66.6)	27 (100)	0.06 ^b^
Yes	6 (20.6)	8 (33.3)	0 (0)	
Currently seeing a psychologist or in the last three months, *n* (%)				
No	20 (68.9)	8 (33.3)	11 (40.7)	0.98 ^b^
Yes	9 (31)	16 (66.6)	16 (59.3)	
Currently seeing a physiotherapist or in the last three months, *n* (%)				
No	9 (31)	8 (33.3)	10 (37)	0.88 ^b^
Yes	20 (68.9)	16 (66.6)	17 (63)	

Abbreviations: LTBCSs: long-term breast cancer survivors, VAS: Visual Analogue Scale, NP: no pain, MP: mild pain, MTSP: moderate to severe pain, *n*: sample size, SD: standard deviation. *p*-values for between-group differences were calculated using the ANOVA test ^a^ and *x*^2^ for categorical variables ^b^.

**Table 2 life-15-00177-t002:** Pain, cancer-related fatigue, physical activity level, and mood state of LTBCSs according to the level of pain in the affected arm.

Variables		LTBCSs’ Level of Pain in the Affected Arm								
		No Pain (NP)	Mild Pain (MP)	Moderate to Severe Pain (MTSP)						
		0–0.99	1–3.99	4–10						
		(VAS)	(VAS)	(VAS)	*p*-Values	COHEN’S *d*	*p*-Values	COHEN’S *d*	*p*-Values	COHEN’S *d*
		(*n* = 29)	(*n* = 24)	(*n* = 27)	NP vs. MP	NP vs. MP	NP vs. MTSP	NP vs. MTSP	MP vs. MTSP	MP vs. MTSP
VAS (cm), mean ± SD, median; IQR, and (95% CI) ^a^										
Non-affected arm		0.30 ± 1.21 0.00; 0.00	0.30 ± 0.57 0.00; 0.75	3.74 ± 3.41 3.65; 7.00	0.07	0	<0.01 **	1.34	<0.01 **	1.39
		(−0.12–0.73)	(0.03–0.56)	(2.38–5.09)						
BPI, mean ± SD, median; IQR, and (95% CI) ^a^										
Interference		0.75 ± 1.70 0.00; 0.64 (0.85–3.25)	2.05 ± 2.56 0.57; 4.46 (0.85–3.25)	3.56 ± 2.94 3.43; 5.29 (2.39–4.72)	0.01 *	0.59	<0.01 **	1.17	0.08	0.54
PFS domains, mean ± SD, median; IQR, and (95% CI) ^a^										
Behavioral/Severity		1.84 ± 2.49 0.33; 3.40 (0.96–2.73)	3.32 ± 2.73 3.66; 6.00 (2.04–4.59)	4.14 ± 2.95 3.50; 5.16 (2.97–5.31)	0.05 *	0.56	<0.01 **	0.84	0.32	0.28
Affective		1.53 ± 2.53 0.00; 1.60 (0.63–2.43)	3.93 ± 2.82 4.40; 5.00 (2.61–5.25)	5.05 ± 3.05 5.80; 4.20 (3.84–6.26)	<0.01 **	0.89	<0.01 **	1.06	0.16	0.38
Sensory		1.69 ± 2.41 0.40; 3.00 (0.83–2.54)	4.40 ± 2.94 5.40; 5.20 (3.02–5.77)	4.97 ± 2.88 5.40; 4.40 (3.83–6.10)	<0.01 **	1.00	<0.01 **	1.23	0.57	0.19
Cognitive/Mood		1.77 ± 2.49 0.50; 2.42 (0.88–2.65)	3.21 ± 2.48 3.25; 4.80 (2.05–4.37)	4.94 ± 2.92 5.17; 5.00 (3.78–6.10)	0.01 *	0.57	<0.01 **	1.16	0.04 *	0.63
Total fatigue score		1.72 ± 2.37 0.64; 2.78 (0.88–2.56)	3.68 ± 2.44 4.41; 4.34 (2.53–4.82)	4.79 ± 2.70 4.82; 3.78 (3.72–5.86)	<0.01 **	0.81	<0.01 **	1.20	0.22	0.43
PFS (Cut-score type A), (%) ^b^										
No fatigue	0–0.9	17 (58.6)	5 (20.8)	3 (11.1)	<0.01 **	-	<0.01 **	-	0.373	-
Mild	1–3.9	7 (24.1)	6 (25)	7 (25.9)						
Moderate	4–6.9	3 (10.3)	11 (45.8)	11 (40.7)						
Severe	7–10	2 (6.8)	2 (8.3)	6 (22.2)						
PFS (Cut-score type B), (%) ^b^										
No fatigue	0–0.9	17 (58.6)	5 (20.8)	3 (11.1)	0.14	-	<0.01 **	-	0.71	-
Mild	1–2.9	5 (17.2)	5 (29.1)	5 (18.5)						
Moderate	3–5.9	5 (17.2)	7 (29.1)	11 (40.7)						
Severe	6–10	2 (6.8)	7 (20.8)	8 (29.6)						
MLTPA (MET hour/week), *n* (%) ^b^										
Inactive (≤3)		4 (13.7)	6 (25)	11 (40.7)	0.47	-	0.05 *	-	0.52	-
Low active (3.1–7.4)		11 (37.9)	11 (45.8)	10 (37)						
Active (≥7.5)		14 (48.2)	7 (29.1)	6 (22.2)						
EVEA, mean ± SD, median; IQR, and (95% CI) ^a^										
Sadness–depression		1.88 ± 2.63 0.75; 3.38 (0.95–2.82)	2.80 ± 2.42 2.00; 3.88 (1.66–3.93)	4.26 ± 2.47 4.25; 3.00 (3.28–5.24)	0.04 *	0.36	<0.01 **	0.93	0.04 *	0.59
Anxiety		2.25 ± 2.25 1.50; 2.88 (1.45–3.05)	2.52 ± 2.54 1.75; 3.56 (1.33–3.71)	4.42 ± 2.60 4.75; 3.75 (3.39–5.45)	0.76	0.11	<0.01 **	0.89	0.01 *	0.73
Anger–hostility		1.46 ± 2.07 0.50; 2.50 (0.73–2.20)	1.78 ± 1.82 1.25; 2.75 (0.93–2.64)	3.54 ± 2.91 2.75; 4.50 (2.39–4.69)	0.24	0.16	<0.01 **	0.82	0.03 *	0.72
Happiness		5.88 ± 2.54 6.00; 4.13 (4.98–6.78)	5.17 ± 2.25 4.75; 3.94 (4.12–6.22)	6.73 ± 9.90 4.75; 4.00 (2.81–10.64)	0.21	0.29	0.27	0.11	0.86	0.21

Abbreviations: LTBCS Long-term Breast Cancer Survivors, VAS Visual Analog Scale, BPI Brief Pain Inventory, PFS Piper Fatigue Scale, MLTPA Minnesota Leisure Time Physical Activity, MET Metabolic Equivalent Task, EVEA Scale for Mood Assessment, NP No Pain, MP Mild Pain, MTSP Moderate to Severe Pain, CI Confidence interval, *n* Sample size, SD Standard deviation, IQR: inter-quartile range. *p*-values for between-group differences were calculated using the *t* test (Mann–Whitney U test) ^a^ for non-normal and the Chi-square ^b^ test for categorical variables. Between-group effect sizes were calculated using Cohen’s ^d^ for continuous variables ^a^. * *p* ˂ 0.05. ** *p* < 0.01.

**Table 3 life-15-00177-t003:** Health-related quality of life of LTBCSs according to the level of pain in the affected arm.

	LTBCSs’ Level of Pain in the Affected Arm								
Variables	No Pain (NP)	Mild Pain (MP)	Moderate to Severe Pain (MTSP)						
	0–0.99	1–3.99	4–10						
	(VAS)	(VAS)	(VAS)	*p*-Values	COHEN’S *d*	*p*-Values	COHEN’S *d*	*p*-Values	COHEN’S *d*
	(*n* = 29)	(*n* = 24)	(*n* = 27)	NP vs. MP	NP vs. MP	NP vs. MTSP	NP vs. MTSP	MP vs. MTSP	MP vs. MTSP
Functioning Scales QLQ-C30, mean ± SD, median; IQR, and (95% CI)									
Physical functioning	32.82 ± 17.97 33.33; 0.00 (27.60–38.04)	29.98 ± 10.25 33.33; 0.00 (25.18–34.77)	33.33 ± 27.73 33.33; 33.33 (22.36–44.30)	0.30	0.19	0.73	0.02	0.80	0.16
Role functioning	91.54 ± 10.82 93.33; 13.33 (87.70–95.38)	81.33 ± 23.25 86.67; 18.33 (70.45–92.21)	69.64 ± 21.19 73.33; 26.67 (61.25–78.02)	0.04 *	0.56	<0.01 **	2.85	0.01 *	0.53
Emotional functioning	91.91 ± 16.20 100.00; 16.67 (86.17–97.66)	79.99 ± 29.91 100.00; 33.33 (65.99–93.99)	63.58 ± 34.29 66.67; 66.67 (50.01–77.14)	0.12	0.49	<0.01 **	1.05	0.06	0.51
Cognitive functioning	72.72 ± 29.92 75.00; 41.67 (62.11–83.33)	75.41 ± 17.82 75.00; 31.26 (67.07–83.76)	47.83 ± 32.81 50.00; 58.33 (34.85–60.82)	0.70	0.10	<0.01 **	0.79	<0.01 **	1.04
Social functioning	74.24 ± 28.59 83.33; 50.00 (64.10–84.37)	67.49 ± 22.60 66.67; 33.33 (56.92–78.07)	42.59 ± 30.42 33.33; 50.00 (30.55–54.63)	0.20	0.26	<0.01 **	1.07	<0.01 **	0.93
Symptom Scales QLQ-C30, mean ± SD, median; IQR, and (95% CI)									
Fatigue	55.55 ± 34.59 66.67; 50.00 (41.87–69.24)	76.66 ± 21.89 75.00; 33.33 (66.41–86.91)	83.83 ± 29.60 100.00; 25.00 (73.34–94.33)	0.06	0.72	<0.01 **	0.88	0.03 *	0.27
Nausea and vomiting	21.88 ± 25.53 11.11; 33.33 (12.83–30.93)	39.44 ± 30.68 33.33; 44.45 (25.08–53.81)	52.26 ± 30.78 44.44; 44.45 (40.08–64.44)	0.02 *	0.62	<0.01 **	1.07	0.15	0.42
Pain	5.55 ± 19.83 0.00; 0.00 (−1.47–12.58)	8.33 ± 19.11 0.00; 0.00 (−0.61–17.28)	9.25 ± 15.56 0.00; 16.67 (3.10–15.41)	0.28	0.14	0.19	0.21	0.55	0.05
Single Items QLQ-C30, mean ± SD, median; IQR, and (95% CI)									
Dyspnea	21.21 ± 26.11 16.67; 33.33 (11.95–30.47)	37.50 ± 26.96 33.33; 33.33 (24.87–50.12)	61.11 ± 29.59 66.67; 50.00 (49.40–72.81)	0.01 *	0.61	<0.01 **	1.42	<0.01 **	0.83
Insomnia	10.10 ± 17.64 0.00; 33.33 (3.84–16.35)	21.66 ± 32.93 0.00; 33.33 (6.25–37.08)	40.74 ± 35.00 33.33; 66.67 (26.89–54.58)	0.24	0.43	<0.01 **	1.10	0.04 *	0.56
Appetite loss	43.43 ± 32.79 33.33; 50.01 (31.80–55.06)	49.16 ± 37.25 50.00; 70.83 (31.72–66.60)	59.25 ± 32.46 66.67; 66.67 (46.41–72.10)	0.56	0.16	0.20	0.48	0.36	0.29
Constipation	9.09 ± 22.47 0.00; 0.00 (1.12–17.05)	18.33 ± 31.48 0.00; 33.33 (3.59–33.06)	9.87 ± 24.13 0.00; 0.00 (0.32–19.42)	0.17	0.34	0.32	0.03	0.22	0.30
Diarrhea	12.62 ± 23.20 0.00; 33.33 (4.39–20.85)	23.33 ± 34.37 0.00; 33.33 (7.24–39.41)	41.97 ± 32.80 33.33; 66.67 (28.99–54.95)	0.26	0.37	<0.01 **	1.03	0.03 *	0.55
Financial difficulties	6.06 ± 17.58 0.00; 0.00 (−0.17–12.29)	11.66 ± 24.83 0.00; 25.00 (0.04–23.29)	19.75 ± 26.56 0.00; 33.33 (9.24–30.26)	0.25	0.26	0.02 *	0.61	0.17	0.31
Global Health Status QLQ-C30, mean ± SD, median; IQR, and (95% CI)									
Global health status	17.17 ± 29.01 0.00; 33.33 (6.88–27.45)	18.33 ± 29.56 0.00; 33.33 (4.49–32.17)	33.45 ± 40.19 33.33; 66.67 (17.55–49.35)	0.80	0.04	0.12	0.46	0.15	0.43
Summary Score QLQ-C30, mean ± SD, median; IQR, and (95% CI)									
Summary score	73.50 ± 11.39 75.73; 9.73 (69.45–77.54)	66.13 ± 16.02 69.81; 24.80 (58.63–73.63)	55.91 ± 14.43 59.23; 20 (50.20–61.62)	0.10	0.53	<0.01 **	1.35	0.01 *	0.67
Functional Scales QLQ-BR23, mean ± SD, median; IQR, and (95% CI)									
Body image	76.26 ± 20.31 75.00; 33.33 (69.05–83.46)	60.83 ± 19.13 62.50; 31.25 (51.87–69.78)	50.30 ± 19.40 50.00; 33.34 (42.63–57.98)	0.01 *	0.78	<0.01 **	1.30	0.10	0.55
Sexual functioning	86.36 ± 22.03 100.00; 16.67 (78.55–94.17)	71.66 ± 28.27 75.00; 56.25 (58.43–84.90)	70.06 ± 31.96 83.33; 58.33 (57.41–82.70)	0.02 *	0.58	0.03 *	0.59	0.93	0.05
Sexual enjoyment	23.23 ± 20.38 33.33; 33.33 (16.00–30.46)	21.66 ± 16.31 33.33; 33.33 (14.03–29.29)	20.37 ± 25.45 16.67; 33.33 (10.29–30.44)	0.89	0.09	0.64	0.12	0.47	0.06
Future perspective	33.33 ± 25.00 33.33; 0.00 (24.46–42.19)	31.66 ± 22.87 33.33; 25.00 (20.95–42.37)	30.86 ± 27.62 33.33; 33.33 (19.93–41.78)	0.96	0.07	0.85	0.09	0.69	0.03
Symptom Scales QLQ-BR23, mean ± SD, median; IQR, and (95% CI)									
Systemic therapy side effects	67.67 ± 34.84 66.67; 33.33 (55.32–80.03)	45.00 ± 40.86 33.33; 91.67 (25.87–64.12)	45.67 ± 37.15 33.33; 66.67 (30.98–60.37)	0.05 *	0.60	0.03 *	0.61	0.93	0.02
Breast symptoms	18.43 ± 19.25 14.29; 19.05 (11.60–25.25)	27.85 ± 22.18 23.81; 23.81 (17.47–38.24)	41.62 ± 20.22 28.10; 33.82 (33.62–49.62)	0.06	0.45	<0.01 **	1.17	0.01 *	0.6
Arm symptoms	10.35 ± 14.88 8.33; 16.67 (5.07–15.63)	19.58 ± 19.17 16.67; 16.67 (10.61–28.55)	50.30 ± 27.97 50.00; 41.67 (39.24–61.37)	0.03 *	0.54	<0.01 **	1.78	<0.01 **	1.28
Upset by hair loss	14.14 ± 22.95 0.00; 22.22 (6.00–22.27)	25.55 ± 15.33 33.33; 22.22 (18.37–32.73)	58.02 ± 30.56 44.44; 55.56 (45.93–70.11)	<0.01 **	0.58	<0.01 **	1.62	<0.01 **	1.46

Abbreviations: LTBCSs: Long-term breast cancer survivors, VAS: Visual Analogue Scale, QLQC30: the EORTC Core Quality of Life Questionnaire, QLQBR23: the Breast Cancer-Specific Module, NP: no pain, MP: mild pain, MTSP: moderate to severe pain, CI: confidence interval, *n*: sample size, SD: standard deviation, IQR: inter-quartile range. *p*-values for between-group differences were calculated using the *t* test (Mann–Whitney U test). Between-group effect sizes were calculated using Cohen’s ^d^. * *p* ˂ 0.05. ** *p* < 0.01.

**Table 4 life-15-00177-t004:** Summary of stepwise multiple regression analysis to determine predictors of pain in the affected arm using the Visual Analogue Scale (VAS).

**Model**	**Variables/Predictors**	**β**	**95% CI**	** *t* **	***p*-Values**	**Linear Regression Equation** **Y = a + bX**
Model 1 (r² = 0.497)	Upset by hair loss (QLQ-C30)	0.70	0.04 ± 0.07	8.78	<0.01 **	Pain in the affected arm = 0.43 + (0.05 * Upset by hair loss)
Model 2 (r² = 0.580)	Upset by hair loss (QLQ-C30)	0.58	0.03 ± 0.06	7.35	<0.01 **	Pain in the affected arm = −0.09 + (0.04 * Upset by hair loss) + (0.25 * Affective domain)
	Affective domain (PFS)	0.31	0.12 ± 0.38	3.89	<0.01 **	
Model 3 (r² = 0.614)	Upset by hair loss (QLQ-C30)	0.53	0.03 ± 0.05	6.62	<0.01 **	Pain in the affected arm = −0.40 + (0.04 * Upset by hair loss) + (−0.18 * Affective domain) + (0.01 * Dyspnea)
	Affective domain (PFS)	0.22	0.04 ± 0.32	2.70	<0.01 **	
	Dyspnea (QLQ-C30)	0.22	0.00 ± 0.03	2.60	0.01 *	
Model 4 (r² = 0.646)	Upset by hair loss (QLQ-C30)	0.52	0.03 ± 0.05	6.78	<0.01 **	Pain in the affected arm = 0.31 + (0.04 * Upset by hair loss) + (0.11 * Affective domain) + (0.01 * Dyspnea) + (0.51 * Alcohol consumption)
	Affective domain (PFS)	0.16	−0.04 ± 0.26	1.93	0.05 *	
	Dyspnea (QLQ-C30)	0.22	0.05 ± 0.07	2.79	<0.01 **	
	Alcohol consumption	−0.19	−0.905 ± −0.012	2.61	0.01 *	

Dependent variable: Pain in the affected arm; r^2^ adjusted coefficient of determination, β regression coefficient, *t* coefficient *t*-value. Abbreviations: VAS: Visual Analogue Scale, QLQC-30: the EORTC Core Quality of Life Questionnaire, PFS: Piper Fatigue Scale, CI: confidence interval. Note: For this analysis, the VAS data as the dependent variable were used in their non-categorized version. *p* ˂ 0.05 */*p* < 0.01 **.

## Data Availability

The datasets generated during and/or analyzed during the current study are available from the corresponding author on reasonable request.

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
