# Peer review of "Assessing the Relationship of Different Levels of Pain to the Health Status of Long-Term Breast Cancer Survivors: A Cross-Sectional Study"

_life, 2025, doi:10.3390/life15020177_

Round 1
Reviewer 1 Report
Comments and Suggestions for Authors
Dear authors,
The authors have developed a well-conducted and well-written study with the aim of investigating the relationship between different pain levels in the affected arm and health status in long-term breast cancer survivors (LTBCS) and to identify predictors of pain at this stage of long-term survivorship.
However, I would like to make a few observations before recommending their work for publication.
The authors note: "Participants were categorized into three groups based on previously established cut-off points for pain measures in the affected arm using the visual analogue scale (VAS), expressed in point-scale ranges from 0-10: 0 to 0.99 no pain, 1 to 3.99 mild pain, 4 to 10 moderate/severe pain (Boonstra AM, Schiphorst Preuper HR, Balk GA, Stewart RE (2014) Cut-off points for mild, moderate, and severe pain on the visual analogue scale for pain in patients with chronic musculoskeletal pain. Pain 155:2545–2550).
But in this paper reade : “In conclusion, we found that VAS scores ≤3.4cm corresponded to mild pain-related interference with functioning, scores of 3.5–6.4 to moderate interference, and scores ≥6.5 to severe interference. VAS scores ≤3.4cm were best described as mild pain, 3.5–7.4 as moderate pain, and ≥7.5 as severe pain for patients with chronic musculoskeletal pain. When patients refer to their pain as moderate, this might be an underestimate of the impact of their pain on functioning. A 3-class solution offered the best fit according to 2 models in our latent class analysis, yielding the classes 0.1–3.8cm, 3.9–5.7cm, and 5.8–10cm.”
Author Response
María Figueroa Mayordomo
Department of Physiotherapy, Faculty of Health Sciences, European University of Valencia,
46112 Valencia, Spain
Tel +34-663746655
Editorial Reviewer 1
Life
15 January 2025
Dear Reviewer 1,
Please find below the answers to each of your contributions
“The authors have developed a well-conducted and well-written study with the aim of investigating the relationship between different pain levels in the affected arm and health status in long-term breast cancer survivors (LTBCS) and to identify predictors of pain at this stage of long-term survivorship.
However, I would like to make a few observations before recommending their work for publication”.
Author response:
First of all, we would like to thank you for your words and the time dedicated to the understanding and improvement of this scientific work. In this way, and from here on, all the answers are detailed, individually and by sections, to each of your suggestions or comments.
With this in mind, we believe that, thanks to the reviewer's contributions and suggestions, together with all our responses, this scientific research is now much easier to read and understand.
General comments
“The authors note: "Participants were categorized into three groups based on previously established cut-off points for pain measures in the affected arm using the visual analogue scale (VAS), expressed in point-scale ranges from 0-10: 0 to 0.99 no pain, 1 to 3.99 mild pain, 4 to 10 moderate/severe pain (Boonstra AM, Schiphorst Preuper HR, Balk GA, Stewart RE (2014) Cut-off points for mild, moderate, and severe pain on the visual analogue scale for pain in patients with chronic musculoskeletal pain. Pain 155:2545–2550).
But in this paper reads: In conclusion, we found that VAS scores ≤3.4cm corresponded to mild pain-related interference with functioning, scores of 3.5–6.4 to moderate interference, and scores ≥6.5 to severe interference. VAS scores ≤3.4cm were best described as mild pain, 3.5–7.4 as moderate pain, and ≥7.5 as severe pain for patients with chronic musculoskeletal pain.”
Author response:
We fully agree with this comment and sincerely appreciate the reviewer’s observation. Upon careful re-evaluation, we have identified that the reference cited in the manuscript does not correspond to the specific cut-off points ultimately utilized in our analysis. We extend our apologies for this oversight, which appears to have arisen from an error during the citation process.
The reference has been updated consistently throughout the manuscript to accurately reflect our approach. Additionally, this feedback prompted us to reflect on the clarity with which the rationale for our selected cut-off points was conveyed. The modifications involve splitting the original mild pain group into two distinct categories (no pain and mild pain) and merging the moderate and severe pain groups. This approach aligns with methodologies observed in previous pain studies (XXX) and was implemented to ensure greater transparency and methodological rigor.
The following sections of the manuscript have been revised to incorporate these updates:
Material and Methods:
Design and Participants
“Based on cut-off points from previous studies, which categorized pain as mild (0–3), moderate (4–6), and severe (≥7) [17], we refined the classification to create three homogeneous groups. Specifically, the original mild pain category (0–3) was subdivided into two groups: no pain (0–0.99) and mild pain (1–3.99). Additionally, the moderate and severe pain categories were combined into a single group, labeled "moderate/severe pain", as seen in previous studies [18]. Consequently, participants were categorized into three groups according to their VAS scores: no pain (0–0.99), mild pain (1–3.99), and moderate/severe pain (4–10).”
Discussion:
Limitations
“Firstly, the approach to categorizing participants into no pain, mild pain, and moderate to severe pain was adapted from previously established and accepted cut-off points, with modifications to ensure the creation of homogeneous groups and to include a distinct no pain category for participants scoring less than 1 on the VAS scale [17, 18], although other cut-off values could have modified our results. However, the inclusion of a distinct no pain category for participants scoring less than 1 on the VAS scale adds value by distinguishing those with no pain (0) from those experiencing mild pain (1-3.99), providing a clearer understanding of pain severity distribution.”
References:
- Zelman DC, Dukes E, Brandenburg N, et al (2005) Identification of cut-points for mild, moderate and severe pain due to diabetic peripheral neuropathy. Pain 115:29–36. https://doi.org/10.1016/j.pain.2005.01.028
- Snijders RAH, Brom L, Theunissen M, van den Beuken-van Everdingen MHJ (2023) Update on prevalence of pain in patients with cancer 2022: A systematic literature review and meta-analysis. Cancers (Basel) 15:591. https://doi.org/10.3390/cancers15030591
“When patients refer to their pain as moderate, this might be an underestimate of the impact of their pain on functioning. A 3-class solution offered the best fit according to 2 models in our latent class analysis, yielding the classes 0.1–3.8cm, 3.9–5.7cm, and 5.8–10cm.”
Author response:
Thank you very much for your thoughtful suggestion. Implementing the proposed changes would require re-performing the entire statistical analysis from the beginning. Moreover, adopting the suggested 3-class subgroups would not allow for the creation of three sufficiently homogeneous groups in our sample, which is essential for ensuring the robustness and validity of the subsequent analyses. We greatly value your input and will certainly consider this approach in future investigations.
The author responsible for correspondence is:
María Figueroa Mayordomo
Department of Physiotherapy, Faculty of Health Sciences, European University of Valencia,
46112 Valencia, Spain
Tel +34-663746655
Sincerely,
María Figueroa Mayordomo

Reviewer 2 Report
Comments and Suggestions for Authors
One of the beautiful studies on pain
Large P values in the text should be written as small p.
Median values should be added to the table where non-parameteric tests are used.
Functioning Scales QLQ and Symptom Scales QLQ-C30, Single Items QLQ-C30 and other scales do not show normal distribution, median values should be added to the table.
What kind of transformation was applied to the independent variables (variables that do not show normal distribution) included in the model in Stepwise Multiple Regression Analysis (square root, 1/x, log?). Stepwise Multiple Regression Analysis used is parametric regression analysis.
If the dependent variable is Vas score, is it included in the model as a dependent variable in a 3-categorical structure as given in the text?
The answers for alcohol consumption and hair loss are categorical.
Both continuous and discrete variables were used together in the model, literature support is required for the usage form.
Author Response
María Figueroa Mayordomo
Department of Physiotherapy, Faculty of Health Sciences, European University of Valencia,
46112 Valencia, Spain
Tel +34-663746655
Editorial Reviewer 2
Life
15 January 2025
Dear Reviewer 2,
Please find below the answers to each of your contributions
“One of the beautiful studies on pain”
Author response:
First of all, we would like to thank you for your words and the time dedicated to the understanding and improvement of this scientific work. In this way, and from here on, all the answers are detailed, individually and by sections, to each of your suggestions or comments.
With this in mind, we believe that, thanks to the reviewer's contributions and suggestions, together with all our responses, this scientific research is now much easier to read and understand.
General Comments
“Large P values in the text should be written as small p”.
Author response:
Thank you very much for your suggestion. We have reviewed all the p-values mentioned throughout the document and these have been modified based on your recommendation.
Tables
“Median values should be added to the table where non-parametric tests are used”.
“Functioning Scales QLQ and Symptom Scales QLQ-C30, Single Items QLQ-C30 and other scales do not show normal distribution, median values should be added to the table”.
Author response:
Thank you for your observation. Taking this comment into consideration, we have made the necessary revisions to include the median and interquartile range (IQR) values for all variables assessed using non-parametric tests, as well as for those variables that did not exhibit a normal distribution (See tables 2 and 3). Additionally, and regarding Figure 1, we have considered it more appropriate not to alter the graph and to continue providing the data as mean plus standard deviation, but the median and interquartile range can be found in the table/supplementary material 2, which is the table containing the detailed information of the whole of Figure 1.
Statistical analysis
“What kind of transformation was applied to the independent variables (variables that do not show normal distribution) included in the model in Stepwise Multiple Regression Analysis (square root, 1/x, log?). Stepwise Multiple Regression Analysis used is parametric regression analysis”.
Author response:
Thank you for your comment. We have reviewed and clarified the transformations applied to the independent variables in the multiple regression model. For variables that did not show normality according to the Kolmogorov-Smirnov test, logarithmic and square root transformations were applied to meet the assumptions of linear regression. This information has been incorporated into the statistical methods section. (Lines 246-248).
Material and Methods
Statistical analysis
“Logarithmic and square root transformations were applied to non-normally distributed variables, as assessed by the Kolmogorov-Smirnov test, to meet linear regression assumptions.”
“If the dependent variable is Vas score, is it included in the model as a dependent variable in a 3-categorical structure as given in the text?”
Author response:
Thank you for your suggestion. In order to perform this part of the statistical analysis, the VAS value used was not in its categorical version but in its scale version before being categorized into the cut-off points. For this reason, we have included a note at the end of Figure 2 and Table 4 to clarify this aspect.
“Note: For this analysis, the VAS data as the dependent variable was used in its non-categorized version”.
“The answers for alcohol consumption and hair loss are categorical”.
Author response:
Thank you for your comment. However, according to the guidelines for the QLQ-C30 and BR-23, the outcome values for “upset by hair loss” should be treated as non-categorical variables. This approach aligns with the methodology used in the majority of studies that reference this evaluation tool (Please see table 3).
Regarding alcohol consumption, this variable was treated as ordinal rather than categorical in the statistical analysis, with higher values indicating greater alcohol consumption. Following the correlation and regression analyses, alcohol consumption exhibited a significant negative correlation. This finding suggests a potential relationship between higher pain levels and lower alcohol consumption. To provide greater clarity on this point, we have considered explicitly specifying this distinction in the analysis section. (Lines 220-224).
“With regards to both categorical and ordinal variables: Chi-square tests were used to compare categorical variables. Furthermore, between-group effect sizes for continuous variables were calculated using Cohen’s d: negligible (d = 0 – 0.19), small (d = 0.2 – 0.49), moderate (d = 0.5 – 0.79), large (d = 0.8 – 1.19), and very large (d = ≥ 1.20) [32].”
“Both continuous and discrete variables were used together in the model, literature support is required for the usage form”.
Author response:
After consulting with a statistics specialist, we have added two new references to the manuscript that provide support for the inclusion of both continuous and discrete variables within the same model. These references substantiate the appropriateness of this approach in the context of our analysis. (Line 241).
References added:
- Chambless LE, Dobson AJ, Patterson CC, Raines B (1990) On the use of a logistic risk score in predicting risk of coronary heart disease. Stat Med 9:385–396. https://doi.org/10.1002/sim.4780090410
- Selk L, Gertheiss J (2021) Nonparametric regression and classification with functional, categorical, and mixed covariates. arXiv [statME]. https://doi.org/10.48550/ARXIV.2111.03115
The author responsible for correspondence is:
María Figueroa Mayordomo
Department of Physiotherapy, Faculty of Health Sciences, European University of Valencia,
46112 Valencia, Spain
Tel +34-663746655
Sincerely,
María Figueroa Mayordomo

Reviewer 3 Report
Comments and Suggestions for Authors
Very Respected Authors,
After carefully reading your paper I have few suggestions. The title of your paper, "Assessing the relationship of different levels of pain on the health status of long-term breast cancer survivors," ... suggests that you will be focusing on determining the levels of pain. In the methodology for assessing pain, you used the Visual Analog Scale, which allows for a rating from 0 to 10, lines 112-114. In the results, you have percentages of patients without pain, with mild pain, and with severe pain, lines 215-217. However, the conclusion lacks information on which level of pain corresponds to a decrease in Health-Related Quality of Life. The results of the study need to be in agreement with the aim and conclusion of the paper. It is necessary to revise the conclusion both in the abstract and in the full paper, lines 554-558.
Author Response
María Figueroa Mayordomo
Department of Physiotherapy, Faculty of Health Sciences,
European University of Valencia,
46112 Valencia, Spain
Tel +34-663746655
Editorial Reviewer 3
Life
15 January 2025
Dear Reviewer 3,
Please find below the answers to each of your contributions
Very Respected Authors,
After carefully reading your paper, I have few suggestions.
Author response:
First of all, we would like to thank you for your words and the months dedicated to the understanding and improvement of this scientific work. In this way, and from here on, all the answers are detailed, individually and by sections, to each of your suggestions or comments.
With this in mind, we believe that, thanks to the reviewer's contributions and suggestions, together with all our responses, this scientific research is now much easier to read and understand.
Abstract, Results and Conclusion
The title of your paper, "Assessing the relationship of different levels of pain on the health status of long-term breast cancer survivors," ... suggests that you will be focusing on determining the levels of pain. In the methodology for assessing pain, you used the Visual Analog Scale, which allows for a rating from 0 to 10, lines 112-114. In the results, you have percentages of patients without pain, with mild pain, and with severe pain, lines 215-217. However, the conclusion lacks information on which level of pain corresponds to a decrease in Health-Related Quality of Life.
The results of the study need to be in agreement with the aim and conclusion of the paper. It is necessary to revise the conclusion both in the abstract and in the full paper, lines 554-558.
Author response:
Thank you for your comment. After a thorough revision of the entire document, we have ensured that the information is unified and consistent across all sections of the paper. In particular, we have revised the abstract, discussion and conclusion sections to specify which groups (levels of pain) are included within the 63.75% of LTBCS referenced in the conclusion.
With these revisions, we believe the title and aim of the paper are now better aligned with the results, conclusion, and other sections of the manuscript, thereby enhancing the overall coherence of the document. Please find the specific modification, by sections, below:
Abstract:
“Conclusion: 63.75% of LTBCS continue to experience mild to moderate to severe pain in the affected arm, negatively impacting their physical, mental, and emotional health status with increased pain severity ≥5 years beyond cancer diagnosis. “Upset by hair loss”, CRF “affective domain”, “dyspnea”, and alcohol consumption collectively explain 64.6% of affected-arm pain level in LTBCS.”
Discussion:
“Furthermore, higher pain severity was associated with increased pain in the non-affected arm, pain interference, CRF, mood disturbances, and physical inactivity, as well as a decreased HRQoL. Notably, 64.6% of affected-arm pain level variance was explained by “upset by hair loss”, CRF “affective domain”, “dyspnea”, and alcohol consumption.”
Conclusion:
“In conclusion, even five years post-BC diagnosis, 63.75% of LTBCS continue to ex-perience mild to moderate to severe pain in the affected arm, which negatively impacts their physical, mental, and emotional health status with increasing pain severity. Moreover, this study provides valuable insights into potential predictors of pain, with “upset by hair loss”, CRF “affective domain”, “dyspnea”, and alcohol consumption collectively explaining 64.6% of its variability in LTBCS.”
The author responsible for correspondence is:
María Figueroa Mayordomo
Department of Physiotherapy, Faculty of Health Sciences, European University of Valencia,
46112 Valencia, Spain
Tel +34-663746655
Sincerely,
María Figueroa Mayordomo

Round 2
Reviewer 1 Report
Comments and Suggestions for Authors
After carefully reviewing the manuscript, I would like to congratulate the authors for the scientific rigour, the writing and the original design of the manuscript.
The article presents a correct and comprehensible writing, on an area of knowledge in which relevant results are provided to be taken into account in the clinical practice of the physiotherapy.
The introduction argues well the need for the study, the method used is reproducible.
The statistical analysis is rigorous, obtaining results that are well represented in the tables and figures.
The discussion section is very well structured, presenting recent references with which to discuss the results obtained.
It is recommended that the section on practical implications be expanded. Also, modify the title of this section from "Clinical Implications" to "Practical Implications".
Congratulations to the authors.
Reviewer 2 Report
Comments and Suggestions for Authors
Corrections made by the author are sufficient.
Acceptable